# Process evaluation of the New Interventions for independence in Dementia Study (NIDUS) Family stream randomised controlled trial: protocol

Danielle Laura Wyman ,[1] Laurie Butler,[2] Claudia Cooper,[3] Peter Bright,[2] Sarah Morgan-Trimmer ,[4] Julie Barber[5]

¹Faculty of Science and Engineering, Anglia Ruskin University - Cambridge Campus, Cambridge, UK
²Psychology, Anglia Ruskin University - Cambridge Campus, Cambridge, UK
³Psychiatry of Older Age, University College London, London, UK
⁴Institute of Health Research, University of Exeter Medical School, Exeter, UK
⁵Statistical Science, UCL, London, UK

**Correspondence to**
Danielle Laura Wyman;
dlw136@pgr.aru.ac.uk

## ABSTRACT

**Introduction** New Interventions for independence in Dementia Study (NIDUS)-Family is an Alzheimer's Society funded new manualised, multimodal psychosocial intervention to support people living with dementia (PLWD) to achieve goals that they and their family carers set, towards living as independently and as well as possible at home for longer. This process evaluation will be embedded within the NIDUS-Family Randomised Controlled Trial intervention-arm (n=199), testing how the intervention influences change, as measured by goal attainment. The evaluation will test, refine and develop the NIDUS-Family theoretical model, associated causal assumptions and logic model to identify key mechanisms of impact, implementation and contextual factors influencing the intervention's effectiveness. Findings will inform how the programme is implemented in practice.

**Methods and analysis** The process evaluation will be theory driven and apply a convergent mixed-methods design. Dyads (PLWD and family carer) will be purposively sampled based on high or low Goal Attainment Scaling scores (trial primary outcome). Qualitative interviews with dyads (approx. n=30) and their respective facilitators post-trial will explore their experiences of receiving and delivering the intervention. Interviews will be iteratively thematically analysed. Matching observational quantitative data will be collected concurrently from videorecordings and/or audiorecordings of NIDUS-Family dyad trial sessions. Further quantitative data will be collected through an acceptability questionnaire for all intervention-arm dyads (n=199). Mixed-method integration will use an interactive analysis strategy, considering qualitative and quantitative findings through mixed-method matrix for dyadic level 'case studies', and a joint display for 'population' level analysis and interpretation.

**Ethics and dissemination** Ethical approval was received from Camden & Kings Cross Research Ethics Committee (REC). Study reference: 19/LO/1667. IRAS project ID: 271 363. This work is carried out within the UCL Alzheimer's Society Centre of Excellence (grant 300) for Independence at home, NIDUS programme. Findings will be disseminated through publications and conferences, and as recommendations for the implementation study and strategy.

**Trial registration number** ISRCTN11425138.

## Strengths and limitations of this study

⇒ This evaluation will place people living with dementia and their family carer as experts to inform how New Interventions for independence in Dementia Study (NIDUS)-Family is implemented in practice.
⇒ This evaluation will use a convergent mixed-methods design grounded within a theory-informed logic model and will follow the Medical Research Council process evaluation guidelines.
⇒ The researcher carrying out this process evaluation is independent from the trial, although funded by the NIDUS programme.
⇒ Data collection occurs post-trial, so there may be a time-lag between dyad finishing the trial and data collection, which may affect recall of experiences.
⇒ Qualitative interviews will occur with approximately 15% of dyads from the intervention arm, such that the results may not be generalisable across other dyads.

## INTRODUCTION

Dementia is a syndrome affecting multiple aspects of a person's cognitive function.[1] Currently an estimated 885 000 people are living with dementia in the UK, and this number is predicted to increase to over 1.2 million by 2030.[2] Approximately two-thirds of people with dementia are living in their own homes,[3] and wish to remain doing so as independently as possible.[4]

Occupational and psychosocial therapy-based interventions are recommended by the UK National Institute for Health and Care Excellence to promote well-being and independence for all people with dementia.[5] Muti-component interventions have demonstrated positive outcomes on a range of measures for people living with dementia (PLWD) and their carers.[6] Due to the complex nature of dementia, outcomes that matter most to PLWD and their carers vary between individuals and over time.[6]



Challenging or distressing behaviours—also known as neuropsychiatric symptoms (NPS)—associated with dementia can lead to family carer stress, poor relationships with home care services, poor self-care, home safety risks and increased healthcare costs and are common reasons why PLWD move to a care home.[4] A systematic review of the effectiveness of psychosocial interventions for managing NPS showed given the positive outcomes, individualised behavioural interventions are possibly efficacious in reducing NPS, and that perceived management of NPS may change, resulting in reduced carer distress and overall cost of care.[6] The review also concluded the most promising interventions for managing NPS seem to be individually tailored behavioural interventions, although more evidence and further research is advised.[6] There is a need to establish an evidence base for interventions in improving personalised support, adaptations, independence and quality of life of PLWD.[7]

### The New Interventions for Independence in Dementia Study: Family

The New Interventions for independence in Dementia Study (NIDUS)-Family programme is a new manualised, multimodal psychosocial intervention to support PLWD to live independently at home for as long, and as well as possible. The intervention focuses on behavioural change, and aims to promote living with quality of life, choice, autonomy, dignity and as independently as possible.[8] The trial's primary objectives are to evaluate the effect of NIDUS-Family and routine care, relative to routine care alone at 12-month follow-up, on goal attainment as measured by family-carer rated Goal Attainment Scaling (GAS) scores, and its cost-effectiveness. Secondary outcomes will measure activities of daily living, quality of life, neuropsychiatric behaviours, apathy, anxiety and depression, and service receipt.[8]

The NIDUS-Family intervention is founded on several theoretical principles (figure 1).

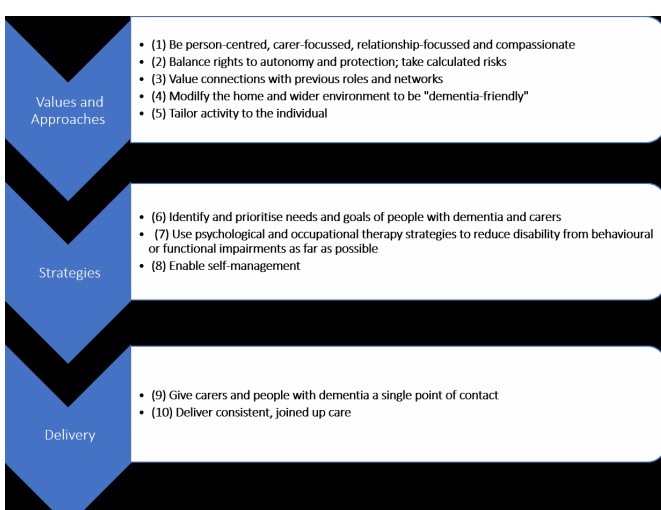

**Figure 1** NIDUS theoretical model of independence at home. NIDUS, New Interventions for independence in Dementia Study.

The NIDUS-Family intervention will recruit 297 participants with a diagnosis of dementia, living in their own home and their regular family carer, who is in at least weekly (including remote) contact. Randomisation will be blocked and stratified by site using a 2:1 intervention: routine care allocation ratio. Consent processes, outcome assessments and intervention delivery will be conducted over 12 months in the participant's own home, at the offices of the recruiting facilitator or via telephone or video call, depending on individual participant preference and COVID-19 restrictions.

The participants randomised to the active intervention group (n=199) will receive between 6 and 8 manualised sessions within the first 6 months. NIDUS-Family aims to support people with dementia and their family carers (a dyad) to address personalised goals aligned to living as well as possible at home.[8] The manualised sessions will be tailored to each participant dyad depending on their preferences and needs and all are delivered by the same facilitator where possible and audiorecorded. The facilitators (graduate psychologists and social researchers with relevant experience but without formal clinical training) will be trained—with 3 days dedicated to research procedures and 9 days to intervention delivery—on how to deliver the intervention and supervised throughout by a clinical psychologist.

Full details relating to the recruitment, design and delivery can be found in the NIDUS study protocol.[8]

The NIDUS-Family intervention can be defined as a complex intervention due to its multiple interacting components, including the personalised and tailored approach, the dyadic relationship and the differing contexts such as living arrangements, within which the programme is implemented.

### Process evaluation of the NIDUS-Family intervention

Process evaluations aim to provide a detailed understanding of an intervention to inform policy and/or implementation into practice. The Medical Research Council (MRC) guidance[9] recommends examining aspects of the intervention including 'implementation (the structures, resources and processes through which delivery is achieved, and the quantity and quality of what is delivered), mechanisms of impact (how the intervention activities, and participants' interactions with them, trigger change), and context (how external factors influence the delivery and functioning of interventions)' (Moore *et al*, p10).[9]

This process evaluation will use the MRC's systematic approach for planning, design, analysis and reporting (online supplemental appendix A shows alignment to the guidance).[9]

### Process evaluation aims and objectives

As recommended by the MRC guidelines, this process evaluation will apply a theory-driven approach to respond to the research question: how does the NIDUS-Family intervention influence goal attainment?

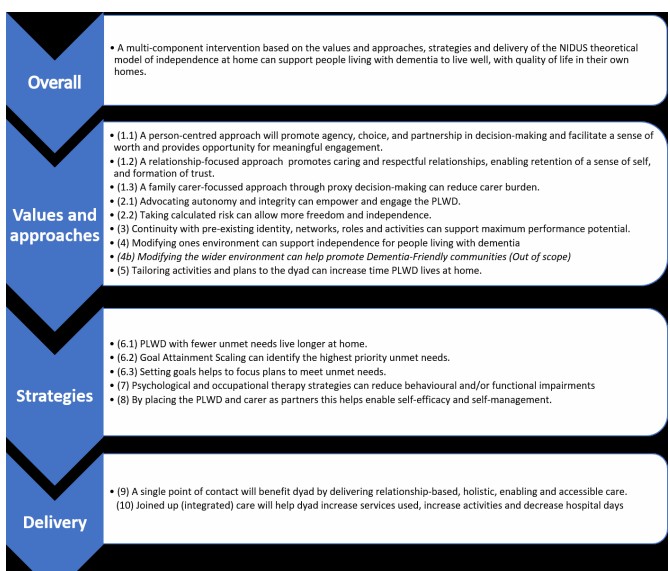

**Figure 2** Hypothesised NIDUS-Family causal assumptions. Derived from NIDUS theoretical model of independence at home (figure 1). NIDUS, New Interventions for independence in Dementia Study; PLWD, people living with dementia.

We will explore how the hypothesised causal chains interact, to test and generate theory about how the NIDUS-Family intervention influences change through:

1. Evaluating the mechanisms of impact, implementation and contextual factors comprising the NIDUS-Family intervention, with a primary focus on factors relating to goal attainment.

2. Identifying which mechanisms, implementation and contextual factors are essential for influencing the effectiveness of the NIDUS-Family intervention.

### Theoretical basis

The NIDUS theoretical model (figure 1), and its underpinning theories[6] directly informed the development of the hypothesised NIDUS-Family intervention causal assumptions (CAs) (figure 2).

The NIDUS-Family logic model (figure 3) in turn clarifies how the NIDUS-Family intervention is designed to realise its intended outcomes and overlays the related CA pathways for goal attainment, values and approaches, strategies, and delivery.

The logic model (figure 3) represents how NIDUS-Family helps dyads to identify 3–5 unmet needs related to living for longer at home. These are turned into specific, measurable, attainable, relevant and time-bound goals. The personalised goals are then mapped onto the NIDUS-Family manualised modules. The dyad attends 6–8 sessions with a facilitator to work through the modules, bringing together old and new strategies to formulate a final action plan for dyads to follow to help

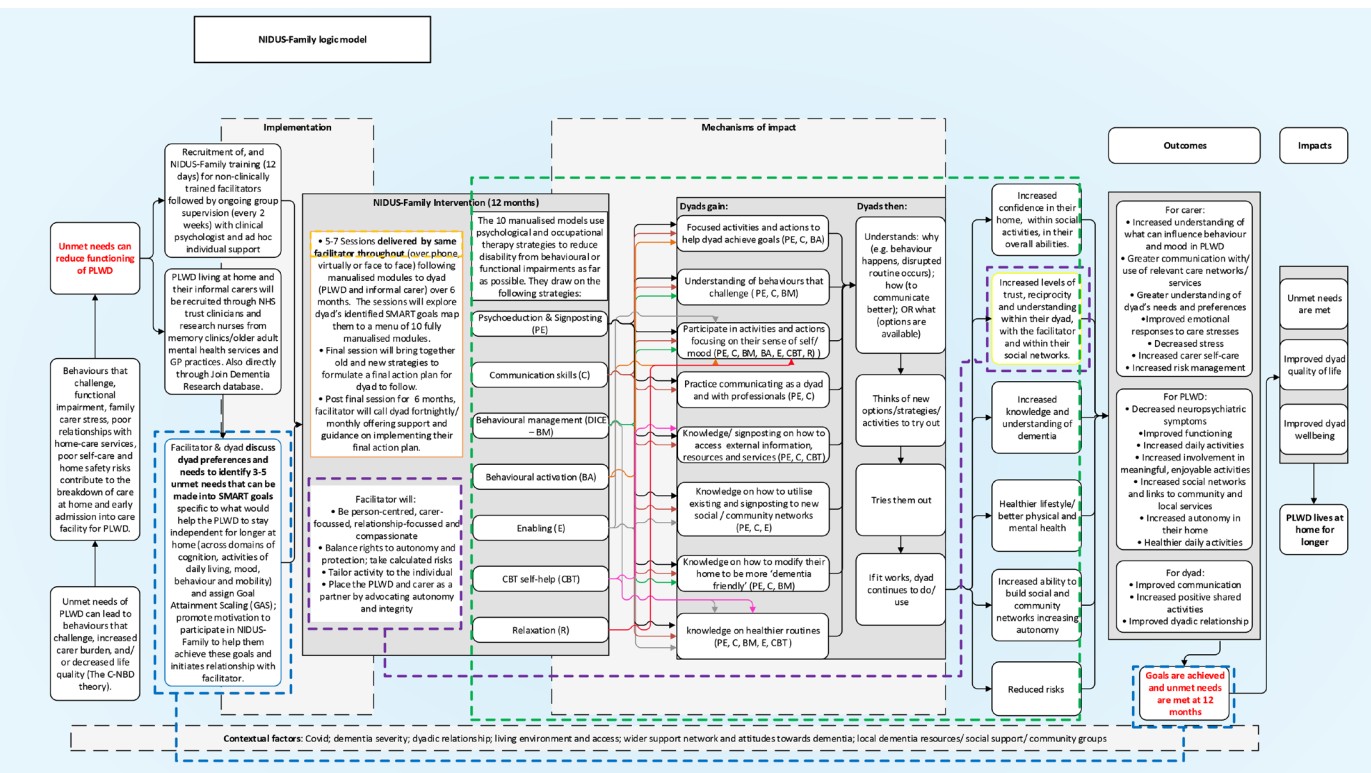

**Figure 3** NIDUS-Family logic model. NIDUS, New Interventions for independence in Dementia Study. Note: Blue section denotes CAs for goal attainment (6.1, 6.2, 6.3), purple section denotes CAs for values and approaches (1.1, 1.2, 1.3, 2.1, 2.2, 5, 8), green section denotes CAs for strategies (3, 4, 7), and yellow section denotes CAs for delivery (9, 10).

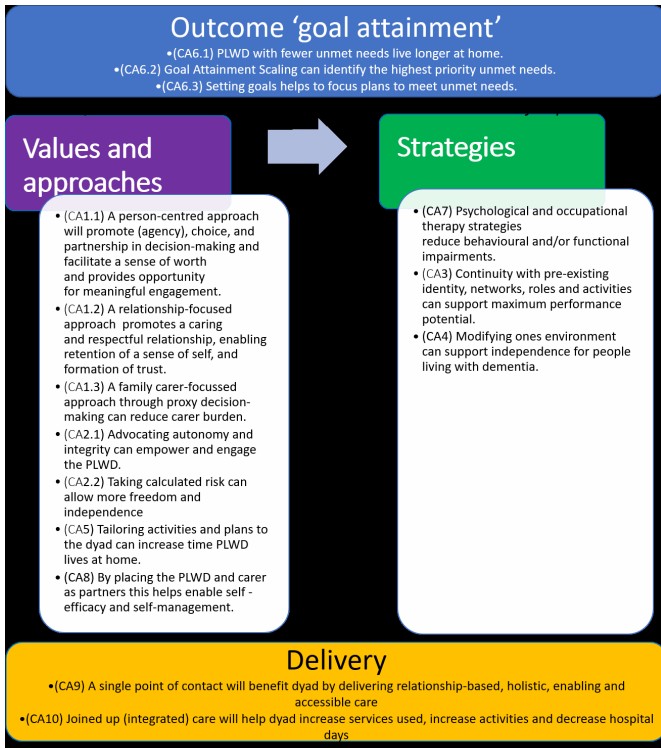

**Figure 4** Emerging theoretical model of change for attainment of dyadic goals through NIDUS-Family intervention (with associated causal assumptions). NIDUS, New Interventions for independence in Dementia Study; PLWD, people living with dementia.

them attain their goals. The intended outcomes for dyads are to improve communication, increase positive shared activities and improve their overall dyadic relationship. By attaining their goals, their unmet needs will now be met, leading to improved quality of life, well-being and the PLWD living at home for longer.

The logic model overlays the hypothesised NIDUS-Family causal pathways (figure 2), the blue pathway represents CAs linked to goal attainment, the purple pathway to values and approaches, the green pathway to strategies, and the yellow pathway represents CAs associated with delivery. Overlaying the CAs onto the logic model details how NIDUS-Family works based on theory and highlights the key pathways that are intended to influence change and will form the focus of this process evaluation.

This process evaluation will evaluate how the pathways delineated in the logic model (figure 3) work in practice. It will test the emerging theory of change for attainment of dyadic goals (figure 4) which represents how the core theoretical principles and casual assumptions derived from the logic model influence behavioural change through goal attainment and posits 'NIDUS-Family supports dyads to attain their goals through applying values and approaches, and strategies, supported by delivery through a single point of contact, and consistent joined up care.' Focusing on dyads with high (2) and low (0 or below) 12-month carer-rated GAS scores will

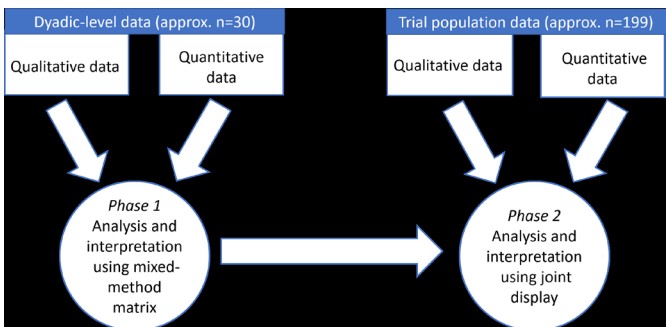

**Figure 5** Mixed-methods convergent design.

identify how the intervention works and enable theory development and refinement.

## METHODS
### Design
A pragmatic paradigm—creating shared meaning and joint action[10]—will underpin the methodology to understand how the NIDUS-Family intervention works. To test, explore, refine and develop the emerging NIDUS-Family theory of change (figure 4) for dyadic goal attainment, and associated hypothesised CAs, a convergent mixed-methods design will be applied (figure 5).[9 11] This design will integrate qualitative and quantitative data—dyadic-level data will be triangulated using a mixed-method matrix to help identify trends and patterns, and population-level data will be integrated with the dyadic-level findings using a joint-display (see section 2.7)—to help gain a more complete understanding of how NIDUS-Family, if effective, influences behavioural, lifestyle and environmental change to enable goal attainment. Qualitative and quantitative questions will be matched on the emerging theory of change constructs for values and approaches, strategies and delivery through a lens of goal attainment (online supplemental appendix B shows matched constructs).[12]

### Patient and public involvement
NIDUS-Family intervention stakeholders (NIDUS clinical psychologists, statisticians, facilitators and the programme manager) and the patient and public involvement group (eight Alzheimer's Society Research Network Volunteers) were consulted in the development of the NIDUS-Family logic model. Consultation occurred in various stages via presentation to map out how the NIDUS-Trial intends to work based on the theory. Feedback was captured and inputted to create the logic model.

### Sampling
For dyadic-level data, dyads will be purposively sampled using quantitative primary measure trial data for 12-month follow-up family-carer rated GAS scores. Dyads with high goal attainment (a score of +2), and dyads with low goal attainment (a score of 0 or below) scores will be invited to interview and their recorded trial sessions (minimum one where available) will be observed through watching/listening to recordings. Dyads' respective facilitators will

be invited to interview. To ensure sufficient conceptual depth, the conceptual depth scale[13] based on range, complexity, subtlety, resonance and validity, will be applied to guide sample size for the number of dyads to be interviewed (approx. 10% N=30). Facilitators will be invited to a second interview when sufficient conceptual depth for dyads is reached to capture any data for subsequently sampled dyads they facilitated. Sampling for high and low 12-month carer-rated GAS scores will help us to understand what influences high scoring dyads to attain their goals, and why low scoring dyads do not attain their goals. This will help us to explore and identify the causal factors which contribute to people benefiting from the intervention.

For trial population-level data, all carers from the intervention-arm will be sent/ invited to complete via telephone call, an acceptability questionnaire at 12-month follow-up. Relevant trial data for dose, reach and attrition will be collected for all intervention-arm dyads. Data will be extracted from the final locked trial database after completion and analysed descriptively. This will help us to understand how the NIDUS-Family intervention influences goal attainment at the trial population level. Fidelity checklists will be applied to a sample of 20% of the intervention-arms transcribed trial session audiorecordings/videorecordings.

## Consent

Trained NIDUS-Family facilitators will assess capacity to consent and obtain written informed consent from each family carer and PLWD prior to NIDUS-Family trial participation. Family carers of people who lack capacity to consent will be asked to complete a consultee declaration form on behalf of their relative with dementia.

Family carers and PLWD will be asked at their 12-month follow-up if they consent to being contacted by the process evaluation researcher. Those that give consent will be contacted and invited to interview to talk about their experiences of receiving NIDUS-Family as part of this process evaluation study. Where the PLWD lacks capacity, interviews will take place with the carer only.

## Data collection

For dyadic-level data collection (intervention-arm dyads sampled for high (+2) and low (0 or below) carer-rated GAS scores at 12-month follow-up) we will collect:

▶ Qualitative semistructured Interviews. Purposively sampled dyads and their facilitator will be invited to separate interviews. Dyad interviews and facilitator interviews will be audiorecorded, anonymised with pseudonyms, transcribed verbatim and uploaded onto NVivo V.12 to manage the data analysis process.
– The dyad semistructured qualitative interview (see online supplemental appendix C for topic guide) will explore their experiences of how the NIDUS-Family approaches and values, and strategies influenced them in attaining their goals for high GAS scores (2+), and why low GAS scores (0 or below) had no change or did not attain their goals.
– The semistructured, qualitative interview with facilitators (see online supplemental appendix D for topic guide) will explore key factors they feel influenced dyadic goal attainment for the dyad(s) selected to whom they delivered the intervention, as well as their overall experiences of facilitating NIDUS-Family. Any novel data relevant to subsequent sampled dyads will be explored with the respective facilitator in a second interview when sufficient conceptual depth for dyads has been captured.

▶ Observational data for purposively sampled dyads attending interview. The evaluator will listen to the dyads' recorded trial sessions (minimum one session per dyad where available). Qualitative (aligned to 'free-text' sections) and quantitative (aligned to Likert scale ratings) data relating to the emerging theory will be captured in an observation checklist (online supplemental appendix E). To ensure validity, a second researcher will independently complete the observation checklist for a minimum of 10% of observed sessions and these observations may be drawn on in the facilitator and dyad interviews. NIDUS-facilitators' session notes will be reviewed to further understand how the values, approaches and strategies were applied for specific dyads.

▶ Quantitative Trial data. Demographic and baseline and 12-month follow-up main trial secondary measure data (facilitator GAS scores at 12 months, functional independence by Disability Assessment for Dementia scale, fidelity checklist data, quality of life for PLWD by DEMQoL or proxy and carer by CarerQoL, NPS by Neuropsychiatric Inventory, Family carer anxiety and depression by Hospital Anxiety and Depression Scale, Apathy of PLWD by The Brief Dimensional Apathy Scale and services used by Client Services Receipt Inventory) related to sampled dyads will be extracted from the trial database and used to describe the dyads included in the qualitative interviews. These data will not be statistically analysed.

▶ Researcher's reflexive field notes will be used to provide in-depth personal perspectives at the level of the dyad.

For trial-population level data, we will collect the following from all intervention-arm participants (n=199):
▶ Family carers will be invited to complete an 'acceptability' questionnaire (online supplemental appendix F) at 12-month follow-up, in which they will rate the extent to which their experiences of the intervention aligned with the core theoretical principles of the emerging theory of change.
▶ We will record trial data for dose (number of sessions), reach (sites and participant location), attrition (number of participants withdrawn) with measures summarised using appropriate tables and graphs.

We will also collect data for trial fidelity (adherence to manualised modules across a sample of 20% of intervention-arm dyads), and withdrawal data where possible from dyads who withdraw. Those who withdraw will either be sent a questionnaire with open questions (online supplemental appendix G) or invited to interview to capture/discuss the reasons for withdrawal (approx. 30 min). Observations (where available) for their sessions can be carried out to identify factors against the observation checklist (online supplemental appendix E). If the dyad is unable to complete the withdrawal questionnaire, their facilitator will be asked to provide information regarding the reasons for withdrawal.

Data will be collected from August 2021 to May 2023 by the process evaluation lead researcher, a postgraduate student who has extensive experience carrying out qualitative interviews with PLWD.

### Data analysis
Dyadic-level qualitative analysis. The qualitative dyadic interviews, facilitator interviews, qualitative observational data and relevant facilitator field notes will be iteratively thematically analysed based on Braun and Clarke[14 15] six phases of thematic analysis, to identify and analyse repeated patterns of meaning. Reflexive field notes will be triangulated with the findings to add depth and further insight.

Dyadic-level quantitative analysis. For observational data inter-rater reliability will be evaluated using the percentage of agreement and the kappa statistic. The quantitative observational checklist ratings for purposively sampled dyads will be used descriptively through tabulating numbers with percentages in each Likert category (strongly agree through to strongly disagree) for each item.

Baseline and 12-month secondary measure trial data and demographic characteristics will be extracted from the final trial database for the purposively sampled interviewed dyads and used descriptively with measures summarised using appropriate tables and graphs.

Trial Population-level qualitative analysis. Acceptability qualitative (free text) data will be analysed thematically. These data will be used to understand convergence or divergence against matched constructs from dyadic-level findings.

Qualitative withdrawal data will be analysed thematically to identify patterns and themes and to better understand the reasons for withdrawing.

Trial Population-level quantitative analysis. Acceptability questionnaire ratings will be reported using descriptive statistics through tabulating numbers with percentages in each Likert category (strongly agree through to strongly disagree) for each item.

We will use summary statistics and graphs to describe participant locations (reach), number of sessions received (dose) and fidelity of delivery to manualised modules. We will report number (%) who withdraw from the intervention/study (attrition rate) and summarise characteristics of those who withdrew against those who did not. These data will be used to evaluate session numbers, geographical distribution and who withdraws.

In a future quantitative study, after main study effectiveness analyses are complete, we will explore whether the number of sessions attended and acceptability scores are associated with intervention effectiveness as defined by 12-month carer-rated GAS scores; as well as exploring how these may differ between carers by sociodemographic characteristics, to understand who NIDUS-Family works for. Analyses will involve fitting multiple regression models including adjustments for confounding factors.

### Qualitative and quantitative integration
A two-phased integration approach will be used to merge and interpret the findings. Phase 1 will integrate data at a dyadic-level and phase 2 will integrate dyadic-level findings with trial population-level data.

Dyadic-level data integration. A mixed-methods matrix will be used to triangulate all dyadic-level qualitative findings (dyad and facilitator interviews, observational data, and field notes) and quantitative data (secondary trial measures, observation checklist data), allowing the data to be openly, actively and interactively considered in the context of each other[16] and 'talk to each other'.[12] These data will be integrated to provide 'case-studies' (approx. N=30). Data will be summarised and displayed in a mixed-method matrix based on the meta-matrix.[17] This will enable analysis between types of data for a single dyad, and identification of patterns across all dyads. The matrix will be used to draw inferences and interpretations of how elements of the intervention interact at the level of the dyad through case-studies and subgroups of high and low GAS score. This will help to explore factors that affect dyads' experiences of, and any benefit from, the intervention helping to gain understanding of how NIDUS-Family influences change in goal attainment at a dyadic level. Generation of themes and patterns will be used to test, develop, and refine the emerging theory that the values and approaches are important, in combination with the strategies, in the success of the intervention for dyads.

Trial Population-level data integration. A joint display[18] will be used to integrate the findings from the dyadic-level matrix with the trial population quantitative and qualitative acceptability questionnaire outcomes and trial data for dose, reach, fidelity and attrition data. This will be used to draw inferences and interpret how NIDUS-Family works at a 'population' level. This approach will enable mapping of dyadic-level data to trial population data to identify the essential mechanisms of impact, implementation and contextual factors that influence change, in turn, refining and consolidating the emerging theoretical model of change for dyadic goal attainment and the NIDUS-Family logic model.

## ETHICS AND DISSEMINATION

The NIDUS-Family trial which is funded by The Alzheimer's Society, has been registered on the clinical trials register at http://www.isrctn.com/ISRCTN11425138. NIDUS-Family ethics, which cover this evaluation have been approved with REC reference: 19/LO/1667, IRAS project ID: 271 363.

The NIDUS study protocol includes a process evaluation section, an amendment was authorised in June 2021 to add additional qualitative interview processes with added facilitator and patient information sheets and consent forms, and to clarify that facilitators will also be interviewed. Changes to the acceptability questionnaire at 12-month follow-up were also submitted and accepted.

Full details relating to ethics can be found in the NIDUS Study protocol.[8]

The evaluation findings will be disseminated through publications and conferences. They will also inform recommendations for a future planned NIDUS-Family implementation study and strategy.

## DISCUSSION

This protocol follows the MRC guidance for process evaluations and outlines the rationale, design and methods for the process evaluation of the NIDUS-Family intervention. The focus of this evaluation is to identify the mechanisms of impact, implementation and contextual factors that influence goal attainment, and how these can help people with dementia live at home for longer. This evaluation is theory-driven and will test, develop and refine the NIDUS-Family theory for goal attainment. Findings will be written up as recommendations that will feed into the NIDUS-Family implementation study.

Study strengths. Following the MRC Guidance—for complex interventions provides a framework for planning, designing, conducting, analysing and reporting a process evaluation. This framework provides a standardised approach to process evaluation enabling transparency across the relatively novel field of complex intervention process evaluation.

A convergent mixed-method design will combine qualitative and quantitative data iteratively to increase understanding of outcomes and improve the NIDUS-Family intervention for roll-out.[9 11] At interpretation level, the two-phase approach initially uses a mixed-method matrix to triangulate qualitative and quantitative data at the dyadic level, then a joint display to integrate dyadic-level findings with trial population data. Each level will draw out new insights and interpretations leading to a deeper understanding of how NIDUS-Family influences change through goal attainment.[19] The mixed-methods design will allow for the emerging theory to be tested, refined and developed to elicit the key mechanisms of impact, implementation and contextual factors that influence goal attainment through NIDUS-Family. These findings will be used to inform the implementation study rolling out NIDUS-Family into practice to maximise its impact in the 'real world'.

The evaluator is independent from the trial, although funded by the NIDUS Programme. To enable understanding and awareness of the trial the evaluator has access to the NIDUS programme team, enabling effective communication and responsiveness to process changes. The evaluation outcomes will not feed into the ongoing trial, they will however inform the post-trial implementation study.

As the NIDUS-Family trial is implemented across various geographical locations, with emphasis on recruiting a diverse population, this reduces the presence of cohort effect.

The evaluator will keep a reflexive journal throughout to capture methodological and theoretical decisions and, transparency and reflection on data collection, analysis and integration.

Study limitations. There are some limitations to this protocol, first, data will only be collected at 12-month follow-up, for some dyads there may be a lag between finishing the intervention and evaluation.

Second, as the NIDUS-Family intervention is a complex model, this process evaluation will evaluate a small sample of participants. As there are many contextual factors (dementia severity, COVID-19, local resources, dyadic relationships) the findings are taken at a specific point in time and account for the contexts relating to that specific dyad so may not be generalisable to difference contexts. The evaluation will reflect and represent the geographical and cultural diversity of the NIDUS-Family trial, this may not be fully reflective of the underlying population.

Finally, it is important to note that even though the facilitators' data will be anonymised, the facilitators are employed by the trial and are involved in trial data collection.

**Contributors** Protocol written by DLW and it was critically edited for intellectual content by LB (ARU), CC (UCL), PB (ARU), SM-T (Exeter) and JB (UCL).

**Funding** This work was carried out within the UCL Alzheimer's Society Centre of Excellence for Independence at home, NIDUS (New Interventions in Dementia Study) programme (Alzheimer's Society Centre of Excellence grant 330).

**Competing interests** The NIDUS-Family trial is funded by the Alzheimer's Society Centre of Excellence grant 330, which is led by CC, with LB as a coinvestigator. DLW is a PhD student funded by the grant.

**Patient and public involvement** Patients and/or the public were involved in the design, or conduct, or reporting, or dissemination plans of this research. Refer to the Methods section for further details.

**Patient consent for publication** Not applicable.

**Provenance and peer review** Not commissioned; externally peer reviewed.

**ORCID iDs**
Danielle Laura Wyman http://orcid.org/0000-0002-4067-0151
Sarah Morgan-Trimmer http://orcid.org/0000-0001-5226-9595

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
