## [Reviewer comments · BMJ Open]

ARTICLE DETAILS

TITLE (PROVISIONAL)	Process evaluation of the New Interventions for independence in Dementia Study (NIDUS) Family stream randomised controlled trial: Protocol
AUTHORS	Wyman, Danielle; Butler, Laurie; Cooper, Claudia; Bright, Peter; Morgan-Trimmer, Sarah; Barber, Julie

VERSION 1 – REVIEW

REVIEWER	Andrews, Randolph ADM Diagnostics LLC
REVIEW RETURNED	15-Dec-2021

GENERAL COMMENTS	If possible, please mention non-human primate research with regard to high stress environments when you discuss your results (e.g., Sapolosky & Uno)
--

REVIEWER	Birkenhäger-Gillesse, Elizabeth University of Groningen, Elderly care medicine and dementia
REVIEW RETURNED	03-Jan-2022

GENERAL COMMENTS	To my opinion is the information on the process evaluation in the manuscript comprehensive and complete. However no information about te start and end date of the study is presented in the manuscript. The strengths and limitations of the study are presented in the 'abstract' and only part of these are in the 'discussion'. I notice that almost all strengths are listed in the abstract and only one limitation, while also most of the limitations as described in the discussion are important enough to be mentioned. In any case, to my opinion, strengths and limitations should be presented in a more balanced way.
--

REVIEWER	Sprange, Kirsty University of Nottingham, Nottingham Clinical Trials Research Unit
REVIEW RETURNED	21-Jan-2022

GENERAL COMMENTS	Thank you for the opportunity to review this protocol. 1. The abstract refs NRES approval, should this be HRA/REC approval?2. Second paragraph of the introduction need expanding as currently does not explain the need for this intervention. E.g. you mention challenging or distressing behaviours as a factor in PLWD being put into formal care - what evidence is there that these behaviours can be managed successfully at home and being at
---

	home would be more beneficial for PLWD. The next sentence about the govt policy also needs explanation. As before why is it important, need to be clearer what are the benefits for PLWD and their family carers? 3. Second paragraph of the introduction - its stated there are no current interventions similar to NIDUS that demonstrate improvements (there are quite a few) - why is this, that so far no interventions have shown a difference - how is NIDUS different to these interventions/trials? 4. There needs to be more detail on the consent process in this population ie if there is a change in capacity during the trial and therefore how this affects the process evaluation sub-study 5. Consistency in terminology ie facilitators are sometimes called researchers? Also need to include a description of who these facilitators were ie clinical, academic and what training they received to deliver the intervention. Evaluating facilitation, how it is delivered and received is paramount due to individual differences and quality of training. 6. fig 3, the logic model, are very confusing and a little messy with lots of coloured lines overlapping making it difficult to follow. 7. Section 2.3 describing PPI input needs expanding. Who were the stakeholders involved, how many people on the PPI group and what did they contribute to the development of the logic-model? 8. Information on analysis in section 2.1 should be moved to 2.5 to prevent repetition. 9. Section 2.4 data collection - who is doing the interviews? Are they experienced interviewing this population? 10. Under 2.6 there is mention of a meta-matrix? This has not been explained previously?
--	--

VERSION 1 – AUTHOR RESPONSE

Reviewer Comments

Reviewer: 1

Mr. Randolph Andrews, ADM Diagnostics LLC

Comments to the Author:

If possible, please mention non-human primate research with regard to high stress environments when you discuss your results (e.g., Sapolsky & Uno) - add consistent with non-human primate research – concordant.

RESPONSE

Thank you for highlighting this interesting study. The current paper and articles reviewed within focus on human determinants of independent living at home. However, it would be interesting in a future review to focus on what the non-human literature might tell us.

Reviewer: 2

Dr. Elizabeth Birkenhäger-Gillesse, University of Groningen, Laurens care centers

Comments to the Author:

(1) To my opinion is the information on the process evaluation in the manuscript comprehensive and complete. However no information about the start and end date of the study is presented in the manuscript.

RESPONSE

I have added more information regarding start and end dates in section 2.5 which now reads as: "Data will be collected from August 2021 through to May 2023 by the process evaluation lead researcher"

(2) The strengths and limitations of the study are presented in the 'abstract' and only part of these are in the 'discussion'. I notice that almost all strengths are listed in the abstract and only one limitation, while also most of the limitations as described in the discussion are important enough to be mentioned. In any case, to my opinion, strengths and limitations should be presented in a more balanced way.

RESPONSE

Thank you for your comments, I have edited the strengths and limitations in the abstract section to be more balanced. This section now reads as:

STRENGTHS AND LIMITATIONS OF THIS STUDY

- This evaluation will place people living with dementia and their family carer as experts to inform how NIDUS-Family is implemented in practice.
- This evaluation will use a convergent mixed-methods design grounded within a theory-informed logic model and will follow the Medical Research Council process evaluation guidelines.
- The researcher carrying out this process evaluation is independent from the trial, albeit funded by the NIDUS Programme.
- Data collection occurs post trial, so there may be a time-lag between dyad finishing the trial and data collection which may affect recall of experiences.
- Qualitative interviews will occur with approximately 15% of dyads from the intervention-arm, such that the results may not be generalisable across other dyads.

Reviewer: 3

Ms. Kirsty Sprange, University of Nottingham

Comments to the Author:

Thank you for the opportunity to review this protocol.

1. The abstract refs NRES approval, should this be HRA/REC approval?

RESPONSE

Thank you, I have edited this to include the REC reference. It now includes:

“REC reference: 19/LO/1667, IRAS project ID: 271363.”

2. Second paragraph of the introduction need expanding as currently does not explain the need for this intervention. E.g. you mention challenging or distressing behaviours as a factor in PLWD being put into formal care - what evidence is there that these behaviours can be managed successfully at home and being at home would be more beneficial for PLWD.

I have added to the introduction to include reference to the evidence that distressing behaviours can be managed at home:

“Occupational and psychosocial therapy-based interventions are recommended by the UK National Institute for Health and Care Excellence (NICE) to promote wellbeing and independence for all people with dementia (NICE, 2018). Multi-component interventions have demonstrated positive outcomes on a range of measures for PLWD and their carers [6]. Due to the complex nature of dementia, outcomes that matter most to PLWD and their carers vary between individuals and over time[6].

Challenging or distressing behaviours—also known as neuropsychiatric symptoms (NPS)—associated with dementia can lead to family carer stress, poor relationships with home care services, poor self-care, home safety risks and increased health care costs and are common reasons why people living with dementia move to a care home.[4, 5] A systematic review of the effectiveness of psychosocial interventions for managing NPS showed given the positive outcomes, individualised behavioural interventions are possibly efficacious in reducing NPS, and that perceived management of NPS may change, resulting in reduced carer distress, and overall cost of care.[6] The review also concluded the most promising interventions for managing NPS seem to be individually tailored behavioural interventions, although more evidence and further research is advised.[6] There is a need to establish an evidence base for interventions in improving personalised support, adaptations, independence, and quality of life of people living with dementia[7].”

3.The next sentence about the govt policy also needs explanation. As before why is it important, need to be clearer what are the benefits for PLWD and their family carers?

RESPONSE

I have expanded this section to include evidence around how tailored interventions can increase time living at home for PLWD, as well as more information around the rising economical cost pushing government policy:

“Introduction.

Government policy emphasises that remaining at home can benefit PLWD through greater quality of life, and society by reducing costs[3]—by reducing transitions of people living with dementia into 24-hour care.[2] To help achieve this, it advocates personalised support and adaptations to help people living with dementia to retain their independence.[6]

3. Second paragraph of the introduction - its stated there are no current interventions similar to NIDUS that demonstrate improvements (there are quite a few) - why is this, that so far no interventions have shown a difference - how is NIDUS different to these interventions/trials?

RESPONSE

Thank you for this insight, I have edited this to include a systematic review of non-pharmacological

interventions . The edited introduction section now reads:

“A systematic review of the effectiveness of psychosocial interventions for managing NPS showed given the positive outcomes, individualised behavioural interventions are possibly efficacious in reducing NPS, and that perceived management of NPS may change, resulting in reduced carer distress, and overall cost of care.[6] There is a need to establish an evidence base for interventions in improving personalised support, adaptations, independence, and quality of life of people living with dementia[7].”

4. There needs to be more detail on the consent process in this population ie if there is a change in capacity during the trial and therefore how this affects the process evaluation sub-study

RESPONSE

Thank you for highlighting this, I have added a section 2.4 which now covers the consent process.

“2.4 Consent

Trained NIDUS-Family facilitators will assess capacity to consent and obtain written informed consent from each family carer and PLWD prior to NIDUS-Family trial participation. Family carers of people who lack capacity to consent will be asked to complete a consultee declaration form on behalf of their relative with dementia.

Family carers and PLWD will be asked at their 12-month follow up if they consent to being contacted by the process evaluation researcher. Those that give consent will be contacted and invited to interview to talk about their experiences of receiving NIDUS-Family as part of this process evaluation study. Where the PLWD lacks capacity, interviews will take place with the carer only.”

5. Consistency in terminology ie facilitators are sometimes called researchers? Also need to include a description of who these facilitators were ie clinical, academic and what training they received to deliver the intervention. Evaluating facilitation, how it is delivered and received is paramount due to individual differences and quality of training.

RESPONSE

Thank you for highlighting this. These terms refer to separate groups of people. I have added a description of who the facilitators are, and I have added more clarity around who the researcher is.

Edits for facilitators can be found in Paragraph 3 of Section 1.1 of the introduction now reads as:

“The facilitators (graduate psychologists and social researchers with relevant experience but without formal clinical training) will be trained—with three days dedicated to research procedures and nine days to intervention delivery—on how to deliver the intervention and supervised throughout by a clinical psychologist.”

6. fig 3, the logic model, are very confusing and a little messy with lots of coloured lines overlapping making it difficult to follow.

RESPONSE

Thank you for highlighting this. As the process evaluation is evaluating a complex trial, the logic model is necessarily complex as it interlinks all the components. We consider it important to confer

understanding around how the trial works and links with all the causal assumptions. This model will be refined during the evaluation, and this may result in a simpler final version. I have added some more narrative and description around the logic model to help add more clarity which sits in the second paragraph under Figure 3:

“The logic model overlays the hypothesised NIDUS-Family causal pathways (Figure 2), the blue pathway represents causal assumptions linked to goal attainment, the purple pathway to values and approaches, the green pathway to strategies, and the yellow pathway represents causal assumptions associated with delivery. Overlaying the causal assumptions onto the logic model details how NIDUS-Family works based on theory and highlights the key pathways that are intended to influence change and will form the focus of this process evaluation.”

7. Section 2.3 describing PPI input needs expanding. Who were the stakeholders involved, how many people on the PPI group and what did they contribute to the development of the logic-model?

RESPONSE

I have added more information about the PPI group and stakeholders, as well as their contribution:

“2.2 Patient and Public Involvement

NIDUS-Family intervention stakeholders (NIDUS clinical psychologists, statisticians, facilitators and the programme manager) and the Patient and Public Involvement (PPI) group (eight Alzheimer’s Society Research Network Volunteers) were consulted in the development of the NIDUS-Family logic model. Consultation occurred in various stages via presentation to map out how the NIDUS-Trial intends to work based on the theory. Feedback was captured and inputted to create the logic model.”

8. Information on analysis in section 2.1 should be moved to 2.5 to prevent repetition.

RESPONSE

Thank you for highlighting this repetition, I have removed this from this section.

9. Section 2.4 data collection - who is doing the interviews? Are they experienced interviewing this population?

RESPONSE

In relation to the comments referring to the researcher, I have added the following to the final paragraph in section 2.5:

“Data will be collected from August 2021 through to May 2023 by the process evaluation lead researcher, a postgraduate student who has extensive experience carrying out qualitative interviews with PLWD.”

10. Under 2.6 there is mention of a meta-matrix? This has not been explained previously

RESPONSE

This is now mentioned in section 2.1 and signposts to section 2.7 for clarity:

“This design will integrate qualitative and quantitative data—dyadic-level data will be triangulated

using a mixed-method matrix to help identify trends and patterns, and population-level data will be integrated with the dyadic-level findings using a joint-display (see section 2.7)—to help gain a more complete understanding of how NIDUS-Family, if effective, influences behavioural, lifestyle and environmental change to enable goal attainment.”

VERSION 2 – REVIEW

REVIEWER	Sprange, Kirsty University of Nottingham, Nottingham Clinical Trials Research Unit
REVIEW RETURNED	04-Apr-2022
GENERAL COMMENTS	The authors have addressed the reviewer comments.